

# Predator–prey interactions between native brine shrimp *Artemia parthenogenetica* and the alien boatman *Trichocorixa verticalis*: influence of salinity, predator sex, and size, abundance and parasitic status of prey

Vanessa Céspedes, Marta I. Sánchez and Andy J. Green

Department of Wetland Ecology, Doñana Biological Station, Seville, Spain

Corresponding author
Vanessa Céspedes,
vanessa.cespedes@ebd.csic.es

## ABSTRACT

*Trichocorixa verticalis* (*T. verticalis*), native to North America and the Caribbean islands, is an invasive waterboatman species (Corixidae) in the southwest of the Iberian Peninsula. Previous studies in the native range have suggested that predation by *T. verticalis* can regulate the abundance of Anostracan and Cladoceran zooplankton in saline ecosystems, causing increases in phytoplankton through a trophic cascade. In this experimental study, we tested the predator–prey relationship between the native brine shrimp *Artemia parthenogenetica*, and *T. verticalis* from the Odiel salt ponds in SW Spain. In three experiments, we investigated (1) the effects of *Artemia* life stage (metanauplii, juveniles, and adults), (2) abundance (three, six, and 12 adult *Artemia*) and (3) parasitic status (*Artemia* infected with avian cestodes or uninfected) on predation rates by *T. verticalis*. We also considered how predation rates in all three experiments were influenced by the sex of *T. verticalis* and by different salinities (25 and 55 g l$^{-1}$). Experiment 1 showed that predation rates were highest for metanauplii, possibly because their photophilic behavior makes them more prone to predation. In Experiment 2, we found that predation rate was higher for female *T. verticalis* and the higher salinity, although the strength of the sex effect varied between treatments. Experiment 3 showed that *T. verticalis* selectively predated adult *Artemia* infected with cestodes (red in color), as previously reported for predation by avian final hosts. Collectively, these results indicate that *T. verticalis* are important predators in their introduced range, and are likely to reduce the abundance of *Artemia* in more salt ponds as they expand their range, thus increasing phytoplankton abundance through trophic cascades.

## INTRODUCTION

Biological invasions are one of the most important drivers of global change and biodiversity loss (*Vilà et al., 2011*; *Simberloff et al., 2013*), and are of particular concern in

aquatic ecosystems (*Bunn & Arthington, 2002*; *Dudgeon et al., 2006*) with impacts at multiple levels of organization (*Simon & Townsend, 2003*).

*Trichocorixa verticalis* (*T. verticalis*) (Fieber, 1851) is one of the few strictly aquatic insects that can be considered as an "alien" species (*Guareschi et al., 2013*), and is the only alien aquatic Hemipteran in Europe. This corixid (length <5.5 mm) is native to North America and the Caribbean islands (*Tones & Hammer, 1975*; *Tones, 1977*; *Kelts, 1979*; *Wurtsbaugh & Berry, 1990*), where it can be the dominant Corixidae species in saline wetlands (*Wurtsbaugh, 1992*; *Aiken & Malatestinic, 1995*) or even survive in the open sea (*Hutchinson, 1931*; *Gunter & Christmas, 1959*) owing to its high osmoregulatory ability (*Scudder, 1976*). The subspecies *T. verticalis verticalis* is now introduced to northwest Africa and the southwestern Iberian Peninsula, where it is often the dominant corixid in saline and hypersaline wetlands (*Carbonell et al., 2017*; *Guareschi et al., 2013*; *Rodríguez-Pérez et al., 2009*). Its success at high salinities is related to its osmoregulatory ability at the egg, nymph, and adult stages; its high fecundity (*Carbonell et al., 2016*); and the release from parasitism by water mites that infect *T. verticalis* at lower salinities (*Sánchez et al., 2015*).

In the native range, *T. verticalis* has been found to be an important predator in saline ecosystems, with the potential to cause major changes through trophic cascades, being one of few predators able to survive in highly mineralized aquatic ecosystems (*Wurtsbaugh, 1992*; *Simonis, 2013*). In the hypersaline Great Salt Lake (USA), the densities of *T. verticalis* and *Artemia franciscana* (Branchiopoda, Anostraca) were negatively correlated, and microcosm experiments indicated that predation by *T. verticalis* reduces the abundance of *Artemia* nauplii resulting in an increase in phytoplankton abundance (*Wurtsbaugh, 1992*; *Wurtsbaugh & Berry, 1990*). A similar trophic cascade effect of *T. verticalis* was observed more recently in a mesocosm experiment in which *T. verticalis* was predating on cladocerans (*Moina macrocopa* and *Daphnia pulex*), releasing phytoplankton from grazing (*Simonis, 2013*). The strength of the cascade caused by *T. verticalis* adults was stronger than that produced by nymphs, which are smaller and less effective as predators (*Simonis, 2013*). The sex of adult *T. verticalis* might also be expected to influence predation rates, since females are larger with higher energetic expenditure (females are 8% longer, see Results). Salinity variation may also influence predation efficiency, especially as *T. verticalis* and *Artemia* overlap in the field at the extreme upper end of the salinity range tolerated by *T. verticalis* (*Wurtsbaugh, 1992*).

Despite its abundance in saline wetlands, there is little information to date on the diet of *T. verticalis*, or its influence on aquatic ecosystems in the introduced range. Stable isotope studies in brackish wetlands in Doñana, SW Spain indicate that *T. verticalis* has an omnivorous diet with an important periphyton component, and that *T. verticalis* itself is an important prey for fish (*Walton et al., 2015*; *Coccia et al., 2016b*). In temporary wetlands of lower salinity where *T. verticalis* coexists with native corixids of a similar size (*Sigara* spp.), the alien often has a lower trophic niche than native species (*Coccia et al., 2016a*). In coastal salt pond complexes in the Portuguese Algarve and SW Spain, *T. verticalis* is now abundant (*Sánchez, Green & Alejandre, 2006a*) and has the potential to influence the abundance of *Artemia* spp. as recorded in the native range (*Wurtsbaugh,*

*1992*; *Wurtsbaugh & Berry, 1990*). *Trichocorixa verticalis* may have an important influence at the ecosystem level by reducing *Artemia* abundance below natural levels, since there is no evidence that native Corixidae were abundant in these systems, and *Artemia* have a strong effect on turbidity and planktonic abundance in these areas (*Sánchez et al., 2016a*). Furthermore, *T. verticalis* may represent an additional threat native *Artemia*, which are already being excluded from many sites by the alien *A. franciscana* introduced from North America for aquaculture (*Amat et al., 2005*).

We present an experimental study of the predation by alien *T. verticalis* of native *Artemia parthenogenetica* in the Odiel salt ponds, SW Spain. We quantify predation rates of different *Artemia* life stages, as a function of *T. verticalis* sex and salinity. We also ask how predation rates are influenced by parasitism of *Artemia* by cestodes. The understanding of the role of parasites in biological invasions requires consideration of how parasites affect interactions between alien species and native competitors and predators (*Combes, 1996*; *Torchin & Mitchell, 2004*; *Prenter et al., 2004*). Cestodes are very prevalent in native *Artemia* populations, with up to 12 species in the study area that use *Artemia* as intermediate hosts in their complex life cycles, and different aquatic bird species as final hosts (*Georgiev et al., 2005*; *Sánchez et al., 2013*; *Redón et al., 2015b*). Cestode parasites cause a bright red coloration and reduced fecundity in native *Artemia*, as well as influencing their behavior, diet and even their tolerance of contaminants (*Sánchez et al., 2013*; *2016b*; *Redón et al., 2015a*). Infected individuals are more vulnerable to predation by birds owing probably to their red coloration (*Sánchez et al., 2009a*; *2009b*), but there are no previous studies of how infection influences predation by other organisms.

Our objectives in this study were: (1) to determine which life history stages of *A. parthenogenetica* are predated by *T. verticalis* in its introduced range; (2) to analyze the differences in predation rates in relation to predator sex and experimental salinity, and how intake rates depend on prey density; and (3) to determine the influence of infection by cestodes on predation rates. We predicted that: (a) *T. verticalis* would preferentially predate earlier developmental stages (metanauplii and juveniles) compared to adult *A. parthenogenetica*; (b) predation rates would be greater at higher salinities, where *T. verticalis* are more likely to co-occur with *Artemia* in the field; (c) predation rates would be higher in female *T. verticalis* due to their larger size and energetic demands; and (d) *Artemia* infected with cestodes will be predated at a higher rate, as previously recorded for birds.

## METHODS

### Sampling

Invertebrates were collected from the Odiel salt ponds in southwest Spain (see *Sánchez, Green & Castellanos, 2006b* for details) during the autumn of 2013. The Junta de Andalucía provided permission to sample (P10-RNM-6262). Approximately 500 *T. verticalis* adults were collected using a hand net of 250 μm mesh from a pond with 25.1 g l$^{-1}$ salinity (37°22′N, 07°00′W). Approximately 3,000 *A. parthenogenetica* individuals (of different life stages) were collected using a hand net (100 μm mesh)

from a pond of 90.3 g l$^{-1}$ (37°25′N, 06°99′W). All specimens were transported to the laboratory in containers filled with pondwater and submerged macrophytes. Before their use in predation experiments, individuals were kept for two days in 7 l aquaria with filtered water from the collection sites, natural substrate and artificial aeration, and were fed periodically each day (with chironomid larvae for *T. verticalis* and microscopic algae (*Tetraselmis chuii*) for *A. parthenogenetica*) until the beginning of the experiment. These aquaria were maintained in a climatic chamber at a temperature of 20–25 °C and a natural light schedule.

### Experimental design

Two experimental salinities (25 and 55 g l$^{-1}$) were selected based on the salinity range in which *T. verticalis* are recorded under natural conditions, and the range at which they are likely to encounter *Artemia* in the wild (*Wurtsbaugh, 1992*; *Rodríguez-Pérez & Green, 2006*; *Carbonell et al., 2016*). Both solutions were prepared by dissolving marine salt (Ocean Fish; Prodac®, Cittadella, Italy) in deionized water.

We conducted three different experiments (Table 1). In each replicate of each experiment, one *T. verticalis* individual (male: length 4.13 ± 0.001 mm, or female: 4.47 ± 0.002 mm; mean ± SE) was placed with *Artemia* prey in a small container (120 ml capacity) with 80 ml of saline solution, and a small plastic mesh (overall size 60 × 75 mm, mesh 10 mm) gripped easily by the corixid to allow it to rest. All experiments were conducted in a climatic chamber (20 ± 1 °C and 12 h light: 12 h dark photoperiod) with a duration of 24 h. After 24 h, we removed the *T. verticalis* from the container and counted the remaining prey. No *T. verticalis* individual was used more than once. Some *T. verticalis* were found dead (mortality rate = 8.2%) and these replicates were removed before analysis. After the experiment, each *T. verticalis* was measured to the nearest 0.01 mm and sexed by close inspection under the binocular microscope; only males have abdominal dextral asymmetry and a strigil present on the left side. Some experimental replicates were recorded on video to allow detailed observation. All statistical analyses were carried out with Statistica, v.13 (StatSoft, Tulsa, OK, USA). Furthermore, details of each experiment are provided below.

### Experiment 1: predation on different developmental stages of *A. parthenogenetica*

To assess selectivity, *A. parthenogenetica* of three developmental stages namely metanauplii (length, 3.96 ± 0.103 mm; mean ± SE), juveniles (6.42 ± 0.184 mm), and adults (7.77 ± 0.130 mm) were provided simultaneously to one adult *T. verticalis*. Juveniles were identified by the absence of an ovisac. In each container, we placed one predator and four individuals of each stage (a total of 12 prey). This was replicated 140 times (35 male + 35 female *T. verticalis* for each of the two salinities (25 or 55 g l$^{-1}$). This experiment was conducted from 10/30/2013 to 11/04/2013.

For each development stage of *Artemia*, the number consumed were analyzed with generalized linear models (GLMs, with a Poisson error and a log link function) to test for differences in consumption rates according to salinity and *T. verticalis* sex. The number

**Table 1 Differences in number of developmental stages of *Artemia* predated by *T. verticalis* (see Fig. 1), compared with Wilcoxon matched pairs tests.**

|  | Valid *N* | *T* | *Z* | *P* |
|---|---|---|---|---|
| Metanauplii and juveniles | 104 | 733 | 6.476 | <0.001 |
| Metanauplii and adults | 110 | 1,316.5 | 5.177 | <0.001 |
| Juveniles and adults | 101 | 1,983 | 2.007 | 0.045 |

of each life stage consumed in each replicate were also compared using non-parametric Wilcoxon matched pairs tests.

## Experiment 2: how *T. verticalis* predation changes with increasing abundance of adult *A. parthenogenetica*

Three treatments with 2, 6, and 12 adult individuals of *A. parthenogenetica* (length 6.92 ± 0.149 mm; mean ± SE) were applied. Each treatment was replicated 40 times (10 male + 10 female *T. verticalis* for each of the two salinities), making a total of 120 replicates. This experiment was carried out from 11/06/2013 to 11/09/2013. The effects of *T. verticalis* sex, salinity, and number of prey provided (categorical variable of three levels) on number of prey consumed (dependent variable) were analyzed with GLMs with a Poisson error and a log link function.

## Experiment 3: the effects of infection by cestodes on the predation of *A. parthenogenetica*

Predation rates were compared for adult *Artemia* infected (length 7.84 ± 0.184 mm; mean ± SE) and uninfected (7.77 ± 0.130 mm) by the avian cestode *Flamingolepis liguloides*, which has a high prevalence in the study area in autumn (*Sánchez et al., 2013*). Individual *Artemia* were observed under the binocular microscope to check their parasitic status (infected with *F. liguloides*). *F. liguloides* is easily visible through cuticle allowing its identification under the binocular microscope (*Georgiev et al., 2005*). All infected individuals had a bright red color whereas uninfected shrimp remained transparent.

In each replicate, we placed one *T. verticalis* and two infected and two uninfected adult prey (i.e., four prey in total). A total of 130 replicates were carried out from 11/11/2013 to 11/16/2013 (30 male + 35 female *T. verticalis* for each of the two salinities). Differences in the number of infected and uninfected *Artemia* were tested using Wilcoxon matched pair tests.

## RESULTS

*Trichocorixa verticalis* consumed on average 42% of *A. parthenogenetica* individuals of all developmental stages and parasitic states used in the three experiments, including 62% of prey in Experiment 1, 35.6% in Experiment 2, and 28.5% in Experiment 3. Direct observations and video recordings (see Additional Information and Declarations) confirmed that mortality of *A. parthenogenetica* was caused by predation by *T. verticalis*. The prey was always alive when captured by *T. verticalis*, which used its forelegs to grab the prey around the brood sac (for adult *Artemia*) or the foremost abdominal segments (for juveniles and metanauplii). *Trichocorixa verticalis* launched their attacks on prey from

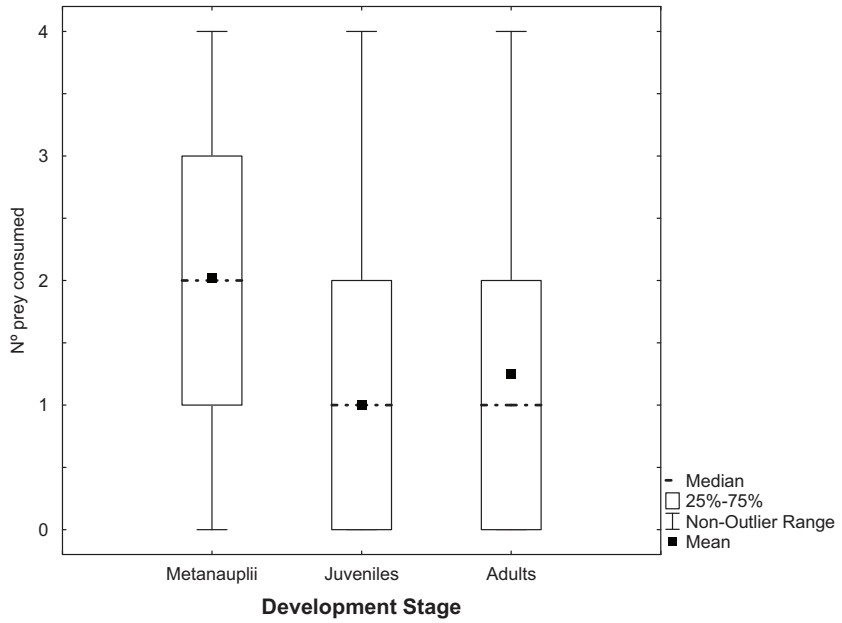

**Figure 1 Predation of *Artemia* according to development stage.** Number of *Artemia* prey of different developmental stages consumed by *T. verticalis* in 24 h (Experiment 1). Shown are range, quartiles, median, and arithmetic mean.

**Table 2 How *T. verticalis* predation rate changes with increasing abundance of adult *Artemia* (see Fig. 2).**

| Effect | Level of effect | Estimate | SE | df | Wald stat. | P |
|---|---|---|---|---|---|---|
| Intercept | | 0.977 | 0.065 | 1 | 227.8606 | <0.001 |
| No. of prey supplied | 2 prey | −0.914 | 0.108 | 2 | 72.0271 | <0.001 |
| | 6 prey | 0.038 | 0.084 | | 0.2028 | 0.652 |
| Salinity | 25 g l$^{-1}$ | −0.186 | 0.051 | 1 | 13.2424 | 0.0003 |
| Sex | Male | −0.177 | 0.065 | 1 | 7.5235 | 0.0061 |
| No. of prey supplied × sex | 6 prey × male | 0.283 | 0.108 | 2 | 6.8780 | 0.008 |
| | 2 prey × female | −0.140 | 0.084 | | 2.7893 | 0.095 |
| Salinity × sex | 25 g l$^{-1}$ × male | −0.117 | 0.051 | 1 | 5.2233 | 0.022 |

Note:
Results of a generalized linear model (GLM with a Poisson error function and log link function) with number of prey (adult *Artemia*) eaten by a single *T. verticalis* as dependent variable, and number of prey provided (2, 6, or 12 prey), *T. verticalis* sex and salinity treatments (25 or 55 g l$^{-1}$) as three categorical predictor variables. The best model selected by AIC is presented. The overall effects of No. of prey supplied ($P < 0.001$) and the interaction No. of prey supplied × sex ($P = 0.029$) were significant. 12 prey, 55 g l$^{-1}$ and female *T. verticalis* are aliased in the model.

the bottom of the experimental container, or when resting on the plastic mesh, then ascended to the water surface after gripping its prey, where it pierced the cuticle and sucked out the soft tissues. Only a small part of the prey's body remained after consumption, then was discarded.

## Experiment 1: predation on different developmental stages of *Artemia*

*Trichocorixa verticalis* predated adults, juveniles and metanauplii of *A. parthenogenetica* (Fig. 1). There were no significant effects in GLMs of *T. verticalis* sex, salinity, or their

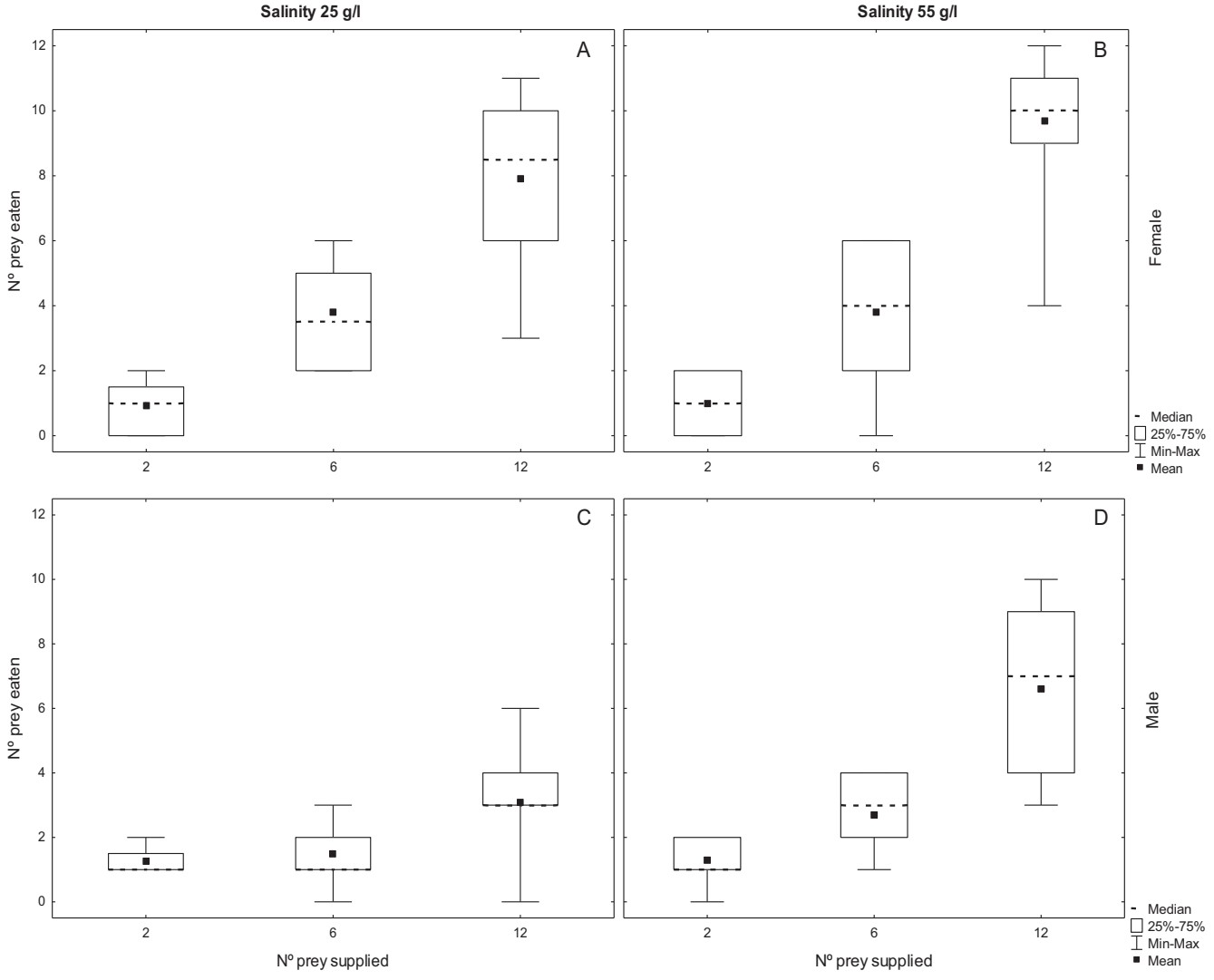

**Figure 2 Predation of adult *Artemia* according to availability.** Number of *A. parthenogenetica* prey consumed by *T. verticalis* in 24 h as a function of number of prey available, *T. verticalis* sex, and salinity (Experiment 2). (A) Salinity 25 g/l (B) Salinity 55 g/l; female. (C) Salinity 25 g/l (D) Salinity 55 g/l; male. Shown are range, quartiles, median, and arithmetic mean.

interaction on the predation rates of each of the three *Artemia* life history stages (Wald tests $P > 0.05$). When sexes and salinity treatments were combined for overall matched-paired tests, the number of metanauplii predated was significantly higher than the number of adults or juveniles (Fig. 1, $P < 0.0001$). Furthermore, the number of adults predated was significantly higher than the number of juveniles ($P = 0.045$).

## Experiment 2: how *T. verticalis* predation changes with increasing abundance of adult *Artemia*

The number of adult *Artemia* consumed by *T. verticalis* increased significantly with the number of prey provided (Table 2). The number predated by females was significantly higher than the number predated by males, whereas the number predated at 55 g l$^{-1}$ was

**Table 3 Determinants of the predation of *Artemia* adults uninfected by cestodes by *T. verticalis* in Experiment 3 (see Fig. 3).**

| Effect | Level of effect | Estimate | SE | df | Wald stat. | P |
|---|---|---|---|---|---|---|
| Intercept | | −0.987 | 0.150 | 1 | 43.175 | <0.001 |
| Salinity | 25 g l$^{-1}$ | 0.351 | 0.150 | 1 | 5.464 | 0.0194 |

**Note:**

Results of a generalized linear model (GLM with a Poisson error function and log link function) with number of uninfected prey eaten by a single *T. verticalis* as dependent variable, *T. verticalis* sex and two salinity treatments (25 and 55 g l$^{-1}$) as two categorical predictor variables. The best model selected by AIC is presented. Salinity of 55 g l$^{-1}$ is aliased. A similar model was conducted for infected prey consumed as dependent variable, and the best model contained only the intercept.

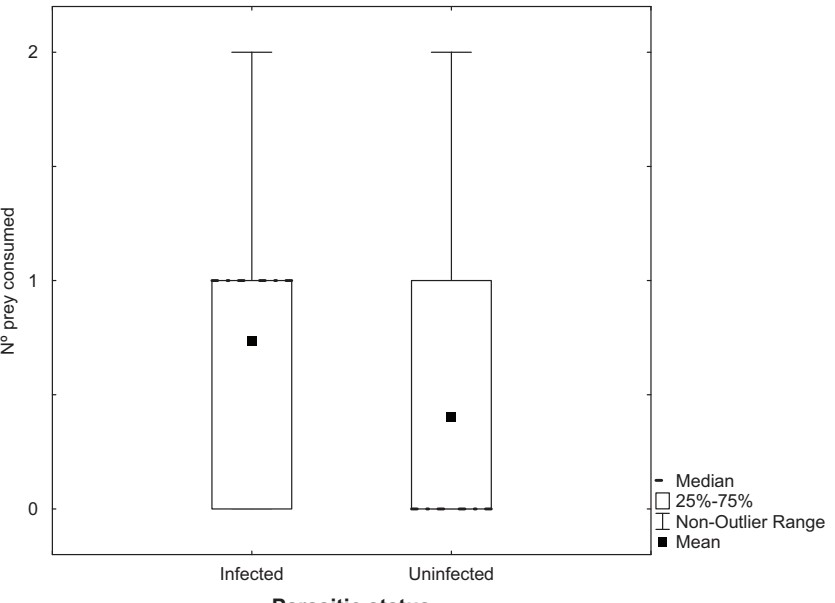

**Figure 3 Predation of adult *Artemia* according to parasitic status.** Number of *Artemia* adult prey consumed (infected or uninfected by cestode parasites) by *T. verticalis* in 24 h (Experiment 3). Results pooling data from different *T. verticalis* sexes and salinities. Shown are range, quartiles, median, and arithmetic mean.

significantly higher than at lower salinity. Furthermore, there was a significant interaction between *T. verticalis* sex and salinity (Table 2). Post hoc tests with Bonferroni correction showed that the increase in predation rate at higher salinity was significant for males but not females (Fig. 2).

## Experiment 3: the effects of infection by cestodes on the predation of *Artemia*

Analysis of the number of *Artemia* adults uninfected with cestodes that were predated by *T. verticalis* with a GLM showed a significant increase at 25 g l$^{-1}$ compared to the higher salinity, but no significant difference between *T. verticalis* sexes (Table 3). A similar model for the number of infected *Artemia* that were predated by *T. verticalis* revealed no influence of *T. verticalis* sex or salinity. The number of infected *Artemia* predated in each

replicate was significantly higher than the number of uninfected individuals (Fig. 3; Wilcoxon matched pairs test, $P = 0.000198$; $N = 129$; $T = 818.5$).

## DISCUSSION

Our results confirm our expectations that *T. verticalis* in the introduced range (subspecies *Trichocorixa verticalis verticalis*) predate brine shrimps, as previously reported for another subspecies (*Trichocorixa verticalis interiores*) in the native range (*Wurtsbaugh, 1992*). We found evidence that predation rates are higher for the larger female *T. verticalis*, and that *T. verticalis* are sensitive to the number and size of their prey, to their parasitic status, and to the environmental salinity.

Our results suggest that *T. verticalis* readily predate *A. parthenogenetica*, as previously shown for *A. franciscana*, with which they coincide in their native range (*Wurtsbaugh, 1992*). In his experiments, *Wurtsbaugh (1992)* found a significant effect of *T. verticalis* on the density of nauplii larvae ($P < 0.01$), but not on adult density ($P = 0.06$). In our experiment, we found that *T. verticalis* had a significant preference for the smallest life stage offered (metanauplii), but they also consumed a high proportion of the adult *A. parthenogenetica*. In addition, we have confirmed in the laboratory that *T. verticalis* readily predate *A. franciscana* adults (V. Céspedes, 2003, personal observation).

Invasive alien species sometimes have an advantage because native prey species may not recognize them as predators (*Sih et al., 2010*). The fact that *T. verticalis* would not encounter *A. parthenogenetica* in their native American range seems to make no difference to their ability to recognize them as suitable prey. Even *T. verticalis* from other habitats in the introduced range which had no prior experience of *Artemia,* seem to instantly recognize native *Artemia* as prey, and feed readily on them in microcosms (V. Céspedes & A. J. Green, 2004, personal observation). Similarly, in a previous experimental study, fish and Odonata larvae predated *T. verticalis* and the native corixid *Sigara lateralis* at a similar rate, although Odonata larvae showed a slight preference for *T. verticalis* as expected from their smaller size (*Coccia, Boyero & Green, 2014*).

*Trichocorixa verticalis* has little difficulty grasping adult *Artemia* which they are able to roll into a ball before they start feeding (see video links in Data Availability). Although we used small experimental containers, increasing the chances of encounters between *T. verticalis* and *Artemia*, *T. verticalis* also actively hunt *Artemia* adults in larger 3.75l microcosms (see video links in Data Availability). *Trichocorixa verticalis* also capture and feed on live benthic chironomid larvae in the laboratory, but with more difficulty as these larvae often wriggle free. In salt ponds, chironomid larvae are an alternative prey item to *Artemia*, and are more abundant in ponds of lower salinity where *Artemia* are rarer (*Sánchez, Green & Alejandre, 2006a*). Copepods are also potential prey items, although it is unclear if they are important in *T. verticalis* diet or not (*Wurtsbaugh, 1992*). *T. verticalis* and the native *Sigara* corixids have similar piercing and sucking mouthparts that allow feeding on soft-bodied invertebrate prey, although they are omnivorous and feed on algae, detritus and periphyton as well as zooplankton and dipteran larvae (*Kelts, 1979*; *Murillo & Recasens, 1986*; *Simonis, 2013*; *Coccia et al., 2016a*)

Brine shrimps and other Anostracans have poor defense mechanisms against predators, and are easy prey for birds, fishes, and aquatic insects. Therefore, Anostracans rely on occupying habitats that are relatively free of predators. Brine shrimps do this by occupying habitats that are too saline for insect predators, whereas fairy shrimps typically do this by occupying temporary aquatic habitats soon after they are flooded and before they are colonized by predators. Hence the addition of a new predator such as *T. verticalis*, which is able to tolerate hypersaline environments, may have a considerable impact on the distribution of *Artemia* in the introduced range.

Our knowledge of the influence of *T. verticalis* on food webs and the abundance of other aquatic organisms in the introduced range is currently very limited, and restricted to a stable isotope study in lower salinity fish ponds and temporary ponds (*Coccia et al., 2016a*), and studies of the niche space occupied by *T. verticalis* and native *Sigara* corixids (*Van De Meutter et al., 2010*; *Carbonell et al., 2016*; *2017*). This is the first study to consider the influence of *T. verticalis* when it invades salt pond systems occupied by *Artemia*. *Artemia* are keystone species and are the most important filter feeders in salt pond ecosystems (*Sánchez et al., 2013*; *2016a*). In the Odiel salt ponds, field studies have shown that *Artemia* are low in abundance or absent in ponds of salinities below 100 g l$^{-1}$ where corixids are present (*Sánchez, Green & Castellanos, 2006b*). Predation by *T. verticalis* and other predators such as the alien fish *Fundulus heteroclitus* is likely to restrict *Artemia* to ponds of higher salinities.

As we predicted, we found that the larger female *T. verticalis* had higher predation rates than males. The evidence we found for prey-size selection has implications for the cascading effects predation has on phytoplankton densities and dynamics (*Simonis, 2013*), since the filter feeding rates of *Artemia* increase strongly with body length (*Sánchez et al., 2016a*). We found significant differences in predation rates between salinities of 55 and 25 g l$^{-1}$, but the results were not very consistent between experiments. In experiment 2, more *Artemia* adults were predated at the higher salinity, this increase being marked more for male *T. verticalis*. In experiment 3, more uninfected *Artemia* adults were predated at the lower salinity, with no salinity effect for infected prey. The lower salinity is closer to the physiological optimum for *T. verticalis* (*Coccia et al., 2013*), which is a highly abundant species in fish ponds in SW Spain of a similar salinity (*Van De Meutter et al., 2010*; *Walton et al., 2015*). It is possible that *T. verticalis* increased *Artemia* predation rates at a higher salinity as a means of compensating for the higher physiological costs of osmoregulation. The change in experimental salinity may also potentially have influenced predation rates through changes in the behavior of the *Artemia* prey (*Sánchez et al., 2009b*), for example, if uninfected prey became less active with a weaker escape response at the lower salinity.

We found that *T. verticalis* were more likely to predate adult *A. parthenogenetica* when they are parasitized by cestodes, as previously recorded for avian predators (*Sánchez et al., 2009a*). This may be connected with the bright red coloration of infected native *Artemia* (*Redón et al., 2015a*), owing to an increased carotenoid content (*Sánchez et al., 2016b*). However, it is unclear whether *T. verticalis* would use color as a cue, as occurs with other Heteroptera (Notonectidae, *Immonen et al., 2014*), and our results may perhaps have been

the product of reduced mobility or escape response of infected *Artemia*. Preference for infected *Artemia* may also be associated with their higher lipid content (*Sánchez et al., 2016b*). Predation of infected individuals is of less significance for the reproductive rate of the *Artemia* population, because cestodes severely reduce the fecundity of infected individuals (*Redón et al., 2015a*; *Sánchez et al., 2016b*). This predation by *T. verticalis* represents a major cost to the cestode parasites since there is no chance of them completing their life cycle.

In conclusion, *T. verticalis* is likely to be important in invaded saline ecosystems owing to its ability to exert top down control on *Artemia* and other prey, causing trophic cascades. In hypersaline systems, the invasion is particularly important since native corixids were absent or rare before the arrival of *T. verticalis*, adding an important predator that is likely to restrict the abundance and distribution of *Artemia*. Future work should compare the functional responses of *T. verticalis* and competing native corixids when feeding on zooplankton and other prey (*Dick et al., 2014*), to clarify the consequences of the invasion on prey communities at lower salinities tolerated by both native and alien corixids. Given the projected expansion of *T. verticalis* over large areas of Europe and the Palaearctic (*Guareschi et al., 2013*) this alien species may have widespread impacts.

## ACKNOWLEDGEMENTS

We are grateful to Raquel López and Anna Badosa for field assistance and to C. Coccia, J. A. Carbonell, A. Millan and J. Velasco for their help and suggestions. The Aquatic Ecology Laboratory (LEA) at EBD-CSIC assisted with the experiments. Research permits were provided by the Marismas de Odiel Natural Park, of the Regional Andalusian Government (Consejería de Medio Ambiente).

### Funding

This research was funded by Fundación Cultural Privada Esteban Romero (Universidad de Murcia, Spain) and project P10-RNM-6262 (Consejería de Innovación, Ciencia y empresa, Junta de Andalucía, Spain). The funders had no role in the study design, data collection and analysis, decision to publish, or preparation of the manuscript.

### Grant Disclosures

The following grant information was disclosed by the authors:
Fundación Cultural Privada Esteban Romero (Universidad de Murcia, Spain): P10-RNM-6262.
Andalusian Project (Consejería de Innovación, Ciencia y empresa, Junta de Andalucía, Spain): P10-RNM-6262.

### Competing Interests

Marta I Sánchez is an Academic Editor for PeerJ.

## Author Contributions

- Vanessa Céspedes conceived and designed the experiments, performed the experiments, analyzed the data, contributed reagents/materials/analysis tools, wrote the paper, prepared figures and/or tables, reviewed drafts of the paper.
- Marta I. Sánchez conceived and designed the experiments, performed the experiments, contributed reagents/materials/analysis tools, wrote the paper, reviewed drafts of the paper.
- Andy J. Green conceived and designed the experiments, analyzed the data, wrote the paper, reviewed drafts of the paper.

## Field Study Permissions

The following information was supplied relating to field study approvals (i.e., approving body and any reference numbers):

Field experiments were approved by Consejería de Medio Ambiente y Ordenación del Territorio (Andalusian, Spain) P10-RNM-6262.

## Data Availability

Dataset on predation:

http://hdl.handle.net/10261/148711

http://dx.doi.org/10.20350/digitalCSIC/8501.

Video links about experiment example (1) and capture of *Artemia* prey by *Trichocorixa verticalis* adult (2):

1) https://www.youtube.com/watch?v=5z2Q5dG33Iw

2) https://www.youtube.com/watch?v=3YutPMRH_PA.

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
