# Peer review of "Predator–prey interactions between native brine shrimp Artemia parthenogenetica and the alien boatman Trichocorixa verticalis: influence of salinity, predator sex, and size, abundance and parasitic status of prey"

_PeerJ, doi:10.7717/peerj.3554_

## Round 0.1 · original submission · Minor Revisions

I have heard back from two reviewers, who both felt your manuscript was generally well done and acceptable to be published in PeerJ after some revisions. Both authors' comments seem fair and constructive, and the scope of their suggested edits is not wide, and therefore my decision is "minor revisions" are needed. I look forward to seeing a revised version of your work.

Reviewer 1 ·

Basic reporting

Some minor English corrections are suggested in the attached review.

Literature refs are sufficient

The figures can use some work. See the attached review.

I did not see a repository listed for the raw data.

Hypotheses and results were appropriate.

Experimental design

Aims and scope of journal met

Research question was well-defined

Investigation was rigorous

Methods can be replicated

Validity of the findings

Findings were valid.

Data is robust, stats appropriate.

Conclusion makes sense.

Additional comments

see attached pdf

Annotated reviews are not available for download in order to protect the identity of reviewers who chose to remain anonymous.

Reviewer 2 ·

Basic reporting

In the paper, authors study the predator-prey relationship between the native brine shrimp Artemia parthenogenetica and Trichocorixa verticalis from the salt ponds in SW Spain, finding an important predator pressure of the alien species that could reduce the abundance of Artemia, leading to trophic effects.

The paper is very well written, the text is clear and unambiguous. English is correct throughout the paper. The structure of the paper is correct and figures and tables relevant and well labeled.

However, I would like to comment some points concerning abstract and introduction section:

1. Line 27. Where you used “infected”, did you mean “uninfected”?
2. Line 48. Change “in press” for “Carbonell et al. 2017”.
3. Line 66. Try to provide a reference for the sentence “females are larger with higher energetic expediture”.
4. Line 78. Cite Wurtsbaugh 1992; Wurtsbaugh and Berry 1990 after “as recorded in the native range”.
5. Lines 97-98. “Infected individuals are more vulnerable to predation by birds”
How? reduced mobility? red colour more attractive?
6. Lines 106-107. In prediction 2 you mentioned that predation rates would be higher at lower salinities, but in introduction you stated that TV and Artemia overlap at the extreme upper end of the salinity range tolerated by TV, and that TV is an important predator in saline ecosystems being able to live in hypersaline waterbodies. Please, clarify this.

Experimental design

Experiments are very well design, described in detail and conducted rigorously and to a high technical standard. I have, though, some comments on this section:

1. Were the individuals of TV starved prior to experiments? it can modify their food intake behavior.
2. Line 140. “each TV was measured”. Level of accuracy?
3. Line 147. Place a comma after “parenthesis” and before “juveniles”.
4. Line 178. I would use 4 infected and 4 uninfected (8 prey in total) preys in the test.

Validity of the findings

Findings are interesting and well discussed. However, I have some comments on results and discussion:

1. Line 185. You did not mention any video recording in Methods section.
2. Lines 214-216. I do not understand this sentence.
3. Line 245. “TV also actively hunt Artemia adults in larger 3.75 l microcosmos” Is this a personal observation?
4. Line 267. Carbonell et al., 2016 & 2017.
5. Line 337-338. Change “in press” for “2017” and “Journal Animal of Conservation” for “Functional Ecology. doi: 10.1111/1365-2435.12884”.

---

## Round 0.2 · accepted · Accept

Your manuscript has been well revised, and is acceptable for publication.

I do have 2 small comments, though, that you should address at the proof stage or earlier if possible.

1. Please use "T. verticalis" and not "TV" throughout the text. As a taxonomist, I believe it is important to use scientific names and not abbreviations. I know a previous reviewer commented on "TV" at the beginning of sentences, but this is incorrect; at the beginning of sentences please use the full genus name.
2. Please ensure there are spaces between the abbreviated generic name and the species name, as in many cases this is missing.

These are minor but essential corrections to an otherwise very well done paper. I look forward to seeing this manuscript in its published form.